# Analysis of Non-Symmetrical Heat Transfers during the Casting of Steel Billets and Slabs

**Adán Ramírez-López [1], Omar Dávila-Maldonado [2], Alfonso Nájera-Bastida [3], Rodolfo D. Morales [2,*], Jafeth Rodríguez-Ávila [4] and Carlos Rodrigo Muñiz-Valdés [4]**

1 Technological and Autonomous Institute of México (ITAM), Department of Industrial Engineering, Rio Hondo #1 Col. Progreso Tizapan, Mexico City CP 01080, Mexico; adan.ramirez@itam.mx
2 Instituto Politécnico Nacional-ESIQIE, Department of Metallurgy and Materials Engineering, Ed. 7 UPALM, Col. Zacatenco, Mexico City CP 07738, Mexico; odavilam@ipn.mx
3 Instituto Politécnico Nacional-UPIIZ, Metallurgical Engineering, Blvd. del Bote 202, Cerro del Gato, Zacatecas CP 98160, Mexico; anajerab@ipn.mx
4 Facultad de Ingeniería, Universidad Autónoma de Coahuila, Blvd. Fundadores Km 13, Ciudad Universitaria, Arteaga Coahuila CP 25350, Mexico; jafethrodriguez@uadec.edu.mx (J.R.-Á.); rodrigo.muniz@uadec.edu.mx (C.R.M.-V.)
* Correspondence: rmorales@ipn.mx; Tel.: +52-1-55-5454-8322

**Abstract:** The current automation of steelmaking processes is capable of complete control through programmed hardware. However, many metallurgical and operating factors, such as heat transfer control, require further studies under industrial conditions. In this context, computer simulation has become a powerful tool for reproducing the effects of industrial constraints on heat transfer. This work reports a computational model to simulate heat removal from billets' strands in the continuous casting process. This model deals with the non-symmetric cooling conditions of a billet caster. These cooling conditions frequently occur due to plugged nozzles in the secondary cooling system (SCS). The model developed simulates the steel thermal behavior for casters with a non-symmetric distribution of the sprays in the SCS using different boundary conditions to show possible heat transfer variations. Finally, the results are compared with actual temperatures from different casters to demonstrate the predictive capacity of this algorithm's approach.

**Keywords:** heat transfer; finite difference method; computer simulation; continuous casting

## 1. Introduction

Steel is one of the essential materials in the world's civilization. It is essential to produce many products such as pipelines, mechanical elements in machines, vehicles, profiles, and beam sections for buildings in many industries. Until the 1950s of the 20th century, steel products required a complex process known as ingot casting; for years, steelmakers focused on developing and simplifying this process. The result was the continuous casting process (CCP); it is the most productive method to produce steel. The CCP allows for producing significant volumes of steel sections without interruption and is more productive than the formal ingot casting process. The CCP begins by transferring the liquid steel from the steel-ladle to a tundish. This tundish or vessel distributes the liquid steel by flowing through its volume to one or more strands having water-cooled copper molds. The mold is the primary cooling system (PCS), solidifying a steel shell to withstand a liquid core and its friction forces with the mold wall.

Further down the mold, the rolls drive the steel section in the secondary cooling system (SCS). Here, the steel section is cooled, solidifying the remaining liquid core by sprays placed in every cooling segment all around the billet and along the curved section of the machine. Finally, the steel strand goes towards a horizontal-straight free-spray zone, losing heat by the radiation mechanism, where the billet cools down further to reach total

solidification. A moving torch cutting scissor splits the billet to the desired length at the end of this heat-radiant zone.

During the CCP, the steel composition, geometrical configuration of the continuous casting machine (CCM), and operating conditions directly affect the billet heat removal and its solidification profiles along the machine's length. To conduct the process under certain operating circumstances, appropriate casting conditions and heat extracting parameters reduce the risks of accidents such as breakouts and other undesirable situations that may affect the production and quality of steel products. Many authors reported simulation results of heat removal conditions during CCP [1–22]. Some of them began using elementary mathematical models using semi-empirical equations to approach heat removal. The computational capacities were limited in the 1960s, 1970s, and 1980s [2,13,20] to addressing the heat transfer problem. The use of numerical methods was not immediately adopted to solve complex problems [12,14–18]. Today, the improvements in data speed processing and the increment in-memory storage allow the software to solve complex problems quickly with complicated calculation routines using substantial data arrays. Moreover, programming methods and techniques provide the user with more efficient tools, algorithms, and friendlier computer environments.

During the CCP of the steel, heat transfer simulation involves calculating the heat removal divided into the three stages (PCS, SCS, and the radiation zone) [1,3–5,7,8,11–19]. Many authors used equations with different coefficients obtained from actual temperature measurements due to the complexity of heat removal in the mold. Savage and Pritchard are the beginners who took measurements from casting molds to predict the heat flux through the mold wall [20]. In the same way, others used interpolation methods to obtain equations representing heat removal conditions in the SCS [1,3–5,7,8]. These equations are general models for the entire SCS. However, some of them developed equations for each segment of the SCS to obtain more accurate results. In this work, the calculation of the heat removal capacity through spray-cooling employs the physical properties of the cooling water [11–19,22–24]. The heat flux in the radiation zone requires a simple calculation as the only equation required is that for radiation. Finally, the model calculates the heat flux inside the steel section, solving the heat conduction equation again to agree with the internal heat re-distribution [25–28].

The temperature profile shown in Figure 1 results after applying uniform heat removal conditions to the four billet surfaces. Here, the isothermal zones are symmetrical in each direction. The geometry of the solidification front goes from quasi-squared-linear faces to the billet corners and surfaces, and then to a circle in the billet core [26,28–31]. Nevertheless, this profile is related to an ideal heat removal which is the most straightforward simplified assumption for the heat removal problem. However, there is a need for differential heat removal conditions on each billet surface due to the fluctuation of the water flow rate and water temperature [6,11,26–29] acting on each surface. Other needs of cooling control is to decrease segregation levels [32], use appropriate heat transfer coefficients [33], use dynamic spray cooling, and use temperature monitoring [34–38]. In addition, many operational problems arise during casting operations, such as a non-homogeneous supply of the water on the billet surface, the boiling heat transfer mechanism of water close to the cold mold face, and the bent or plugging spray nozzles. Thus, defining features and boundary conditions permit special heat removal conditions [39–43].

Different from previous publications [23,24,35,36,42,43], the present work places emphasis on the non-symmetric cooling conditions of billets and slabs leading to non-symmetric temperature profiles inside these sections. This contribution describes the improvements of the mathematical model used in these simulations, characterized by its versatility and readiness for process analysis, process control, and machine design.

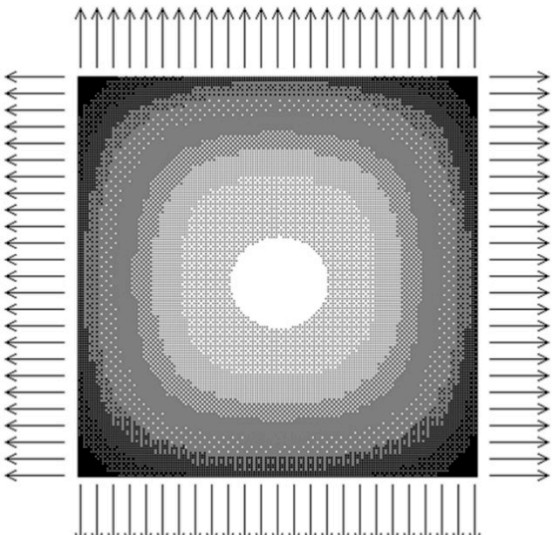

**Figure 1.** A typical temperature profile in the cross section of a steel billet [23].

## 2. Computational Representation of Steel Casting

The computation procedure involves the discretization of the volume elements of a squared steel section using finite difference techniques and the designation of the dimensions of this volume is lx, ly, and lz. The inclusion of each 3D element is an independent node in a squared mesh, stored in a 2D computational array. The calculation of the element sizes ($\Delta x$) and ($\Delta y$) are functions of the billet section and the elements used for discretization. The calculation of the magnitude of the dimension in the cast direction ($\Delta z$) obeys the stability criterion calculated according to the casting speed. The user defines the number of nodes for (Nx) and (Ny). The identification of the array is by the literals in the sub-indexes (I) and (J). The solution of the heat transfer equation is through a finite difference method employing nested loops [23,24,39–43]. The thermo–physical properties, such as thermal conductivity ($k$) and heat capacity ($C_p$) for steel, are functions of the temperature and chemical composition [14–18,22–24].

The following assumptions constitute part of the models applied in this work:

- The casting temperature (TCO) is the same for all the nodes. Thus, the assignment of the energy required to define the casting temperature is for all nodes.
- Only one single steel volume is in the casting plant for the simulation. In consequence, there is no heat inter-change in the longitudinal direction of the machine. Thus, the heat removal in the cast direction is negligible. This assumption simplifies the problem and reduces the calculation time; the problem is a 2D type as the longitudinal heat transfer is negligible. Therefore, the treatment of the problem uses 2D computational arrays; one for enthalpies and one for both the latest and previous calculations of enthalpy ($H_{i,\,j}^{t-1}$ and $T_{i,j}^{t-1}$) and ($H_{i,j}^{t}$ and $T_{i,j}^{t}$), reducing the computer's memory requirements.
- The simulation begins at the meniscus level inside the mold. Then, the simulation time is ($t = 0$).
- The step time ($\Delta t$) is calculated as a function of the billet dimensions (lx) and (ly) using (Nx) and (Ny) nodes, and the steel thermal diffusivity ($\alpha$) is given in Equation (1) where k is the thermal conductivity, $\rho$ is density, and $C_p$ is heat capacity.

$$\alpha = \frac{k}{\rho C_p} \tag{1}$$

The counting of iterations includes the entire algorithm-loop to calculate the heat transfer and corresponding routines for both displaying and saving the information nested inside. Every iteration corresponds to the results obtained after updating the actual

simulation time with the step time ($t = t + \Delta t$). The step time ($\Delta t$) obeys the criterion given by Equation (2) and its estimation is through the casting speed by knowing the dimension ($\Delta z$) of the steel volume control.

$$\Delta t = \frac{\Delta x \Delta y}{4\alpha} \tag{2}$$

The calculation of transformation temperatures for steel (temperature of liquidus, $T_{Liq}$, temperature of solidus, $T_{Sol}$, upper transformation temperature, $T_{AR1}$, and lower transformation temperature, $T_{AR3}$) is through Equations (3)–(6) as a function of the steel chemical composition [22]. Therefore, Equation (7) calculates the corresponding energy required to melt the steel ($H_{i,j}$) [3,15,22]. Here, ($w$) is the weight of each discretized steel element obtained using Equation (8). A graphical representation of the energy calculated for a steel volume is in Figure 2.

$$T_{Liq} = 1537 - 88\%C - 25\%S - 5\%Cu - 8\%Si - 5\%Mn - 2\%Mo - 4\%Ni - 1.5\%Cr - 18\%Ti - 2\%V - 30\%P \tag{3}$$

$$T_{Sol} = 1535 - 200\%C - 12.3\%Si - 6.8\%Mn - 124.5\%P - 183.9\%S - 4.3\%Ni - 1.4\%Cr - 4.1\%Al \tag{4}$$

$$T_{AR1} = 723 - 10.7\%Mn - 16.9\%Ni + 21.9\%Si + 16.9\%Cr + 290\%As + 6.38\%W \tag{5}$$

$$T_{AR3} = 910 - 203\%C - 15.2\%Ni + 44.7\%Si + 104\%V + 31.5\%Mo + 13.1\%W - (30\%Mn + 11\%Cr + 20\%Cu - 700\%P - 400\%Al - 120\%As - 400\%Ti) \tag{6}$$

$$H = q = \int_{T=T_0}^{T=T_{AR1}} Wc_p dT + \int_{T=T_{AR1}}^{T=T_{AR3}} Wc_p dT + \int_{T=T_{AR3}}^{T=T_{Sol}} Wc_p dT + \int_{T=T_{Sol}}^{T=T_{Liq}} Wc_p dT + \int_{T=T_{Liq}}^{T=T_\infty} Wc_p dT \tag{7}$$

$$w = \Delta x \cdot \Delta y \cdot \Delta z \cdot \rho_{steel} \tag{8}$$

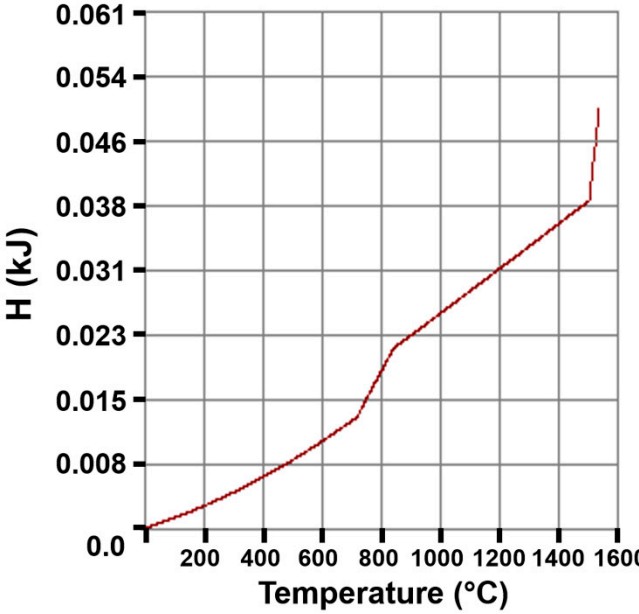

**Figure 2.** Relationship between enthalpy and temperature for a steel billet [23].

Some authors simulated the billet thermal behavior during CCP considering only 1/4 or 1/2 of the cast section and assuming the symmetric heat removal as is described in Figure 3 [1–4,10,13–17]. Heat removal takes place only through the lateral surfaces and there is a heat conduction assumption on the rest of the sample. Formerly, the application of this procedure was limited due to computer capacities at the time.

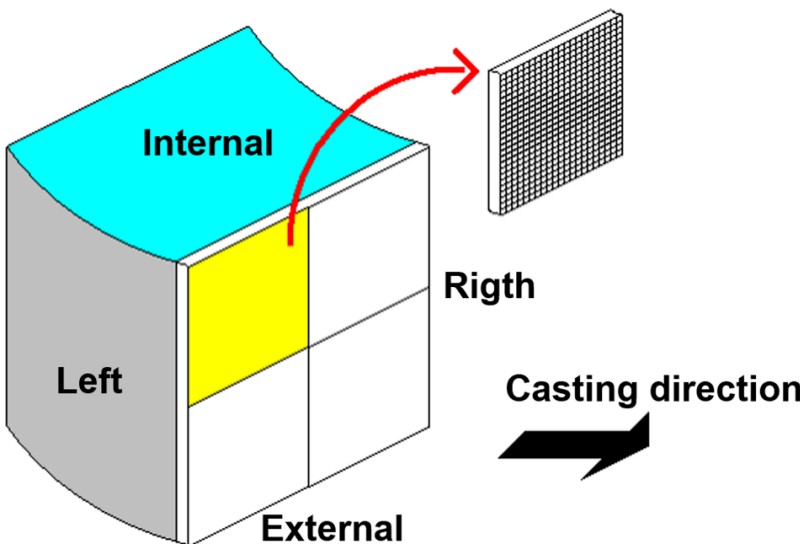

**Figure 3.** Billet faces used in the mold. Internal and external faces refer to the internal and external radius of the machine radius of the curvature. The het flow is symmetric [23].

### 3. Heat Transfer and Conduction inside the Billet Core

The first step during the calculation process is to obtain the temperatures in the external nodes at each step time ($t + \Delta t$). The new value of enthalpy in the external nodes is feasible by using Equation (9). The enthalpy value for every node ($H_{i,j}^t$) is the last and the heat removal is ($q_{i,j}$). Then, the new values are stored in a new array for the next iteration and the previous array is deleted and updated for efficient use of the computational resources. The calculation of the applied heat removal for every node in the billet surfaces ($q_{i,j}$) is a function of the mechanism involved according to the billet position and the CCM [5–7,24–26].

$$H_{i,j}^t = H_{i,j}^{t-1} - q_{i,j} \tag{9}$$

Similarly as presented in Equation (7), for each value of enthalpy, ($H_{i,j}$) corresponds to a value of temperature ($T_{i,j}$) [13,15,26–29]. The main routine updates the heat that remained after each step time ($\Delta t$) during the simulation. If the cast speed is known, it is possible to calculate exactly the time at which each node changes from liquid to a mushy structure and from a mushy structure to a solid-state, as is indicated in Equation (10), after which data is stored in a pair of 2D computational arrays, namely ($t_{sol\,i,j}$ and $t_{liq\,i,j}$). A comparison routine works for this purpose. This routine is applied to all external and internal nodes and is included in the main calculation routine to update the information. Here, the superscripts ($t$) and ($t − 1$) represent the corresponding values of the latest and previous iterations during the simulation time. Computationally, these values correspond to the actual time ($t$) and the previous simulated time ($t − \Delta t$).

$$H_{i,j}^t \to T_{i,j}^t \to t_{sol_{i,j}},\ t_{liq_{i,j}},\ t_{mushy_{i,j}} \tag{10}$$

The best fit for calculating the mold's heat removal is possible by using equations that involve dwelling time, as Savage and Pritchard demonstrated [20]. In the present work, the algorithms use Equation (11) to calculate heat removal using the coefficients calculated by these authors, but the sub-indexes (s) provide the user the option to define different heat removal conditions to each billet surface. Heat removal is considered a complex problem and many authors prefer to treat it by using coefficients as a function of dwell time.

$$q_s = A_{o_s} + B_{o_s}\sqrt{t} \tag{11}$$

Heat removal in the SCS is due to two mechanisms. When steel is under a sprayed zone, the heat removal is intense and the surface temperature decreases due to the forced

convection. Nevertheless, when the steel is running under a non-sprayed area, the heat removal is through the radiation mechanism and the billet surface temperature increases due to the latent heat flux coming out from the core [6,9,21–24]. Equations (12) and (13) calculate the heat removal under sprayed and no sprayed areas, respectively. Here, the heat flux ($q$) is a function of a heat transfer coefficient ($h$), which results from the previous calculation of the water flow applied in the nozzles. This calculation includes the evaluation of the Prandtl, Nusselt, and Reynolds numbers. The water and billet surface temperatures are in the boundary ($T_w$) and ($T_{i,j}$). Equation (13) is the Stefan–Boltzmann law and calculates the heat flux value as a function of the steel emissivity ($\varepsilon$). These two heat removal conditions work during the simulation when the geometrical conditions of the CCM are verified and validated. The result is a temperature curve that goes down when steel is under a sprayed area and goes up when steel is under a non-sprayed area. The entire process to calculate the coefficient ($h$) solves Equations (14)–(17). The sub-indexes "*ns*" and "*side*" indicate that these values can differ for every segment of the SCS and every billet side. The sub-index "*w*" identifies the liquid used as water. ($\varepsilon$) is the emissivity, ($\mu$) is the dynamic water viscosity, and the sub-indexes ($i$) and ($j$) identify the nodal positions of the billet surface.

$$q_s = h_f \left( T_{i,j} - T_w \right) \tag{12}$$

$$q_s = \sigma \varepsilon \left( T_{i,j}^4 - T_{amb}^4 \right) \tag{13}$$

$$Re_{ns,side} = \frac{d_{wns,side} v_{wns,side} \rho_{wns,side}}{\mu} \tag{14}$$

$$Pr_{ns,side} = \frac{c_p \mu}{k} \tag{15}$$

$$Nu_{ns,side} = cRe^n Pr^{0.333} \tag{16}$$

$$h_{ns,side} = \frac{Nu_{ns,side} k}{D} \tag{17}$$

Within the billet core, conduction is the only heat transfer mechanism involved. The heat re-distribution is available by solving the partial differential Equation (18), which explains that a temperature profile exists for the steel volume at each time step of the simulation ($t + \Delta t$). The enthalpy and temperature calculations include the billet's frontal billet face at each step time. The solution of this equation includes a pair of nested loops to calculate each node temperature of the steel volume. Using the information of the nearest neighbors in the previous iteration and solving mathematical Equation (19) through the Crank–Nicholson method [6,9,10] allow for the determination of the temperature field to be feasible.

$$\left( \frac{\partial^2 T}{\partial x^2} + \frac{\partial^2 T}{\partial y^2} \right) = \frac{1}{\alpha} \frac{\partial T}{\partial t} \tag{18}$$

$$\frac{1}{(\Delta x)^2} \left( T_{i-1,j} + T_{i+1,j} - 2T_{i,j} \right) + \frac{1}{(\Delta y)^2} \left( T_{i,j-1} + T_{i,j+1} - 2T_{i,j} \right) = \frac{1}{\alpha_{i,j}} \frac{T_{i,j}^{t+1} - T_{i,j}^t}{\Delta t} \tag{19}$$

In the computing flowchart shown in Figure 4, the need to separate the external and internal nodes of the analyzed mesh is evident. The calculation of the external nodes proceeds in agreement with the steel position and CCM for each time step ($t + \Delta t$).

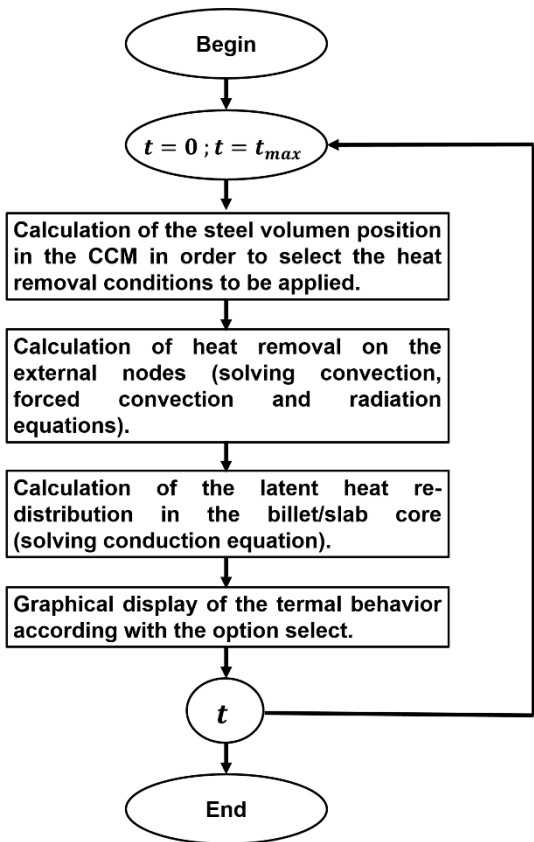

**Figure 4.** Flow chart of the mathematical model.

Storing the information concern the heat removal during the simulation in two-dimensional computational arrays works to feed the information in the solution process into the mold. The data stored for defining the operating conditions have three locations that refer to the coefficients of the Savage and Pritchard equation, which defines the final heat flux removal [20]; then, for the primary cooling system (PCS), i.e., the mold, the array used is:

$$PCS = [Surface], [coefficient \ 1], [coefficient \ 2]$$

For the SCS, a five-dimensional computational array defines all the operating conditions in the SCS. The locations have the following format:

$$SCS = [Segment], [Spray], [Surface], [Lateral \ Position], [Data]$$

The first box refers to the cooling segment with different spray distributions. Then, the second box defines the position in the SCS of the control volume at any time. By comparing it with the running distance, it is possible to know exactly under what spray is the billet element. Then, the appropriated heat removal is applied according to the corresponded surface. More lateral sprays can be defined if necessary. The data file stores all the information required for each spray, such as the water flow rate, temperature, and shooting angle. Segments and sprays are ordered numerically for easy recognition using integers to store the corresponding values. A numerical code identifies the billet surface at which the heat removal is applied. The code employed is used for (1) the external radius side of the billet, (2) the internal radius side of the billet, (3) and the left and (4) right billet surfaces, respectively. The only restriction is that all these values must be larger than zero.

Figure 5a–h show different industrial spray arrangements for a SCS, applied to billet continuous casting machines. Here, four sprays around the billet surfaces are placed at the same arrangement and apply the same water flow rate.

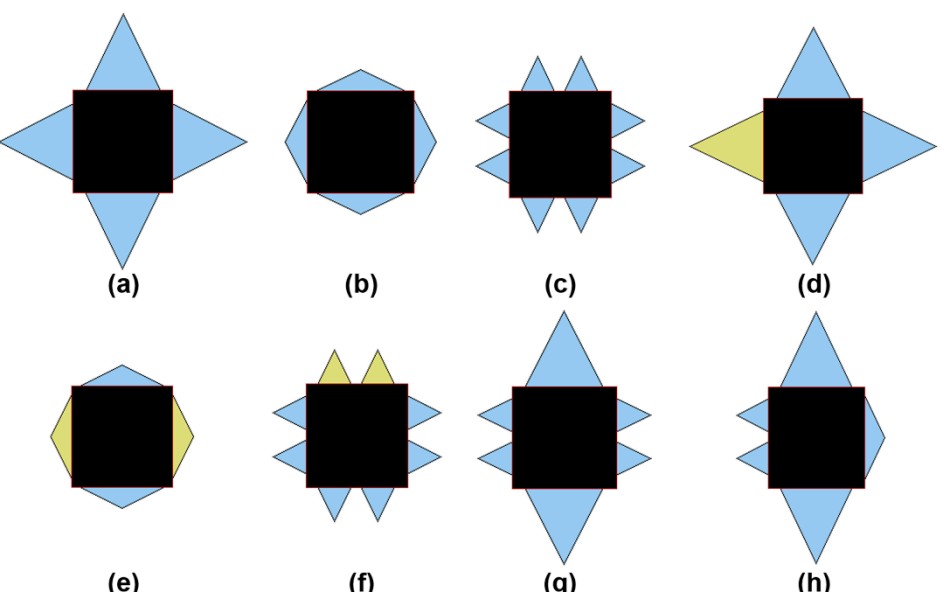

**Figure 5.** (**a**–**h**) Different arrangements of billet cooling sprays in continuous casting machines.

Then, the same heat flux works in the four billet surfaces, resulting in symmetrical temperature profiles. Figure 5b shows a version of the original shooting angle [8,10,16,22–24] and Figure 5c shows another symmetrical spray arrangement but with more than one spray shooting water over the billet surfaces. These figures provide a homogeneous heat removal. Figure 5d–f show the same spray configuration but with different water flow rates over one or two billet surfaces, yielding different heat fluxes. Figure 5g–h show that different spray arrangements for heat removal perform differently for each billet surface. There are different spray configurations during the casting of slabs, especially for quenching internal and external faces, which are the broad faces, as shown in Figure 6.

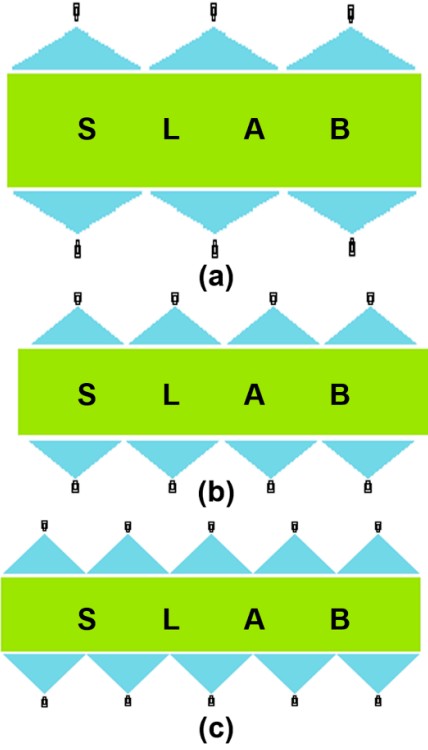

**Figure 6.** (**a**–**c**) Different arrangements of slab cooling sprays in continuous casting machines.

## 4. Process Simulation

### 4.1. Case 1

The information concerning the CCM geometry used as input data for the simulation corresponds to a current caster, including the billet section dimensions, as shown in Table 1. The casting temperature and phase changing temperatures were calculated according to the steel composition shown in Tables 2 and 3. The iterating conditions for each segment of the SCS are in Table 4. Here, the number of sprays is the same in segments (1) and (2) but not in segment (3), as shown in the CCM layout of Figure 7. Moreover, the operating conditions, such as the water flow rate and shooting angle ($\Omega$), are also different for each segment. Consequently, there are no symmetrical heat removal conditions quenching the steel billet surface.

**Table 1.** Casting conditions.

| Cast Speed (m/min) | Casting Temperature (°C) | RC (m) | $\theta_0$ | $l_x$ | $l_y$ |
|---|---|---|---|---|---|
| 2.40 | 1535 | 7.45 | 5.8 | 130 | 130 |

**Table 2.** Temperatures of the steel.

| $T_{ariq}$ (°C) | $T_{sol}$ (°C) | $T_{AR3}$ (°C) | $T_{AR1}$ (°C) |
|---|---|---|---|
| 1524.38 | 1507.85 | 844.07 | 721.04 |

**Table 3.** Steel composition in mass percentage.

| C | Al | Cr | Cu | Mn | Nb | Mo |
|---|---|---|---|---|---|---|
| 0.380 | 0.003 | 0.05 | 0.040 | 1.050 | 0.002 | 0.002 |
| **Ni** | **P** | **Ti** | **S** | **Si** | **Sn** | **V** |
| 0.006 | 0.014 | 0.002 | 0.018 | 0.200 | 0.001 | 0.002 |

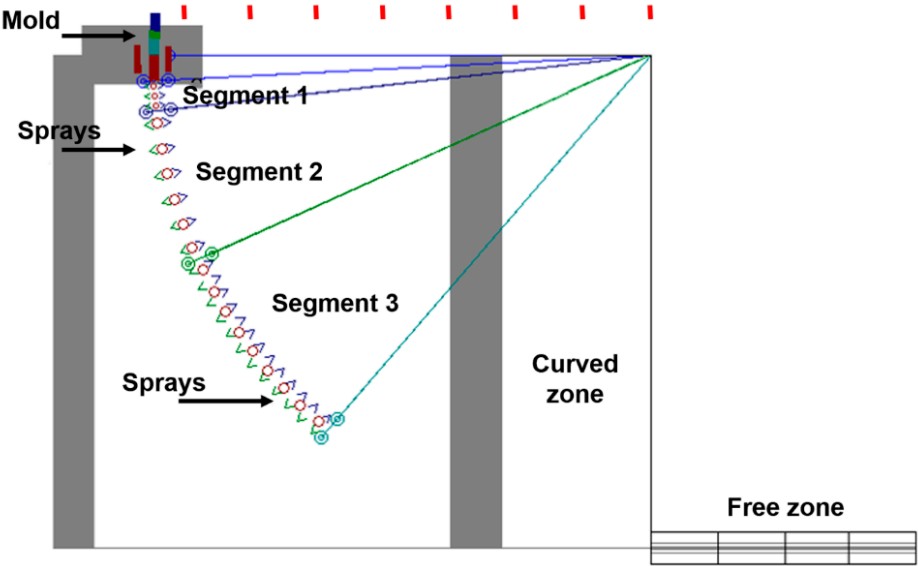

**Figure 7.** Layout of the billet casting machine corresponding to case 1.

**Table 4.** Operating conditions of the SCS (segments 1, 2, and 3). Note: Int. = internal and Ext. = external.

| Segment | 1 | | | | 2 | | | | 3 | | | |
|---|---|---|---|---|---|---|---|---|---|---|---|---|
| Surface | Internal | External | Left | Right | Internal | External | Left | Right | Internal | External | Left | Right |
| Water flow rate (L/min) | 7 | 10 | 7 | 10 | 7 | 10 | 7 | 10 | 7 | 10 | 7 | 10 |
| Sprays on cast direction | 3 | 3 | 3 | 3 | 6 | 6 | 6 | 6 | 13 | 12 | 9 | 9 |
| Sprays on the lateral direction | 2 | 2 | 2 | 2 | 1 | 1 | 1 | 1 | 1 | 1 | 1 | 1 |
| Nozzle diameter (m) | 0.003 | | | | 0.003 | | | | 0.003 | | | |
| $\Omega_{cast\ dir}$ | 50 | | | | 60 | | | | 60 | | | |
| $\Omega_{lateral\ dir}$ | 60 | | | | 50 | | | | 50 | | | |
| $D_{bs}$ (m) | 0.083 | | | | 0.100 | | | | 0.100 | | | |
| $\theta$ | 3 | | | | 17 | | | | 25 | | | |

### 4.1.1. Operating Conditions and Assumptions

The information concerning the CCM geometry used as input data for the simulation corresponds to a current caster, including the billet section dimensions, as shown in Table 1. The casting temperature and phase changing temperatures were calculated according to the steel composition shown in Tables 2 and 3. The iterating conditions for each segment of the SCS are in Table 4. Here, the number of sprays is the same in segments (1) and (2) but not in segment (3), as shown in the CCM layout of Figure 7. Moreover, the operating conditions, such as the water flow rate and shooting angle ($\Omega$), are also different for each segment. Consequently, there are no symmetrical heat removal conditions quenching the steel billet surface.

The simulation takes into count the following further assumptions:

- The steel composition is homogeneous.
- The cast speed is constant during the simulation.
- The heat removal inside the mold is constant and equal on each side of the billet.
- The operating and quenching conditions are constant during the casting operation.

The model developed calculates different solidification rates with intense or weak heat removal to include all probable risks. Moreover, a water supply failure simulation is also possible by defining some segments or sprays with a low-value water flow rate, which is equal to zero if it is absent.

The CCM in Figure 7 and casting conditions described in Tables 1–4 belong to case (1) for analysis and validation purposes, as found in the following lines. Here, (RC) is the curved radius of the CCM, ($\theta$) is the angle of every segment measured as a function of (RC), and ($\theta_0$) is the first angle of the SCS and is measured from the end of the mold. ($\Omega$) is the shooting angle of every spray.

### 4.1.2. Simulations and Results

Figure 8a shows the corresponding surface temperature of each billet surface [23,24]. The simulation includes the curved region of the CCM to appreciate details of the non-symmetrical temperature profiles. The close-up of Figure 8b illustrates the influence of different spray disposals along the casting direction.

The temperature on the billet surfaces became different in the segments of the SCS because of the different heat removal conditions applied. There was the same number of sprays in the first and second segments in the lateral and casting directions but the water flow rates were different. Consequently, different curves diverged at the end of the mold position for these segments, as shown in Figure 8. The other two curves for

the corresponding billet surfaces were behind these because the sprays were at the same distance and with the same heat removal conditions. This condition also evidences the precision of the algorithm, the method, and the effectiveness of the number of the nodes employed. The difference between the other curves is not significant and the curves are hidden or superimposed. The 40,000 nodes used for the simulation generated no significant errors after every step time due to blunders and rounded methods, without a strong influence on the temperature profiles.

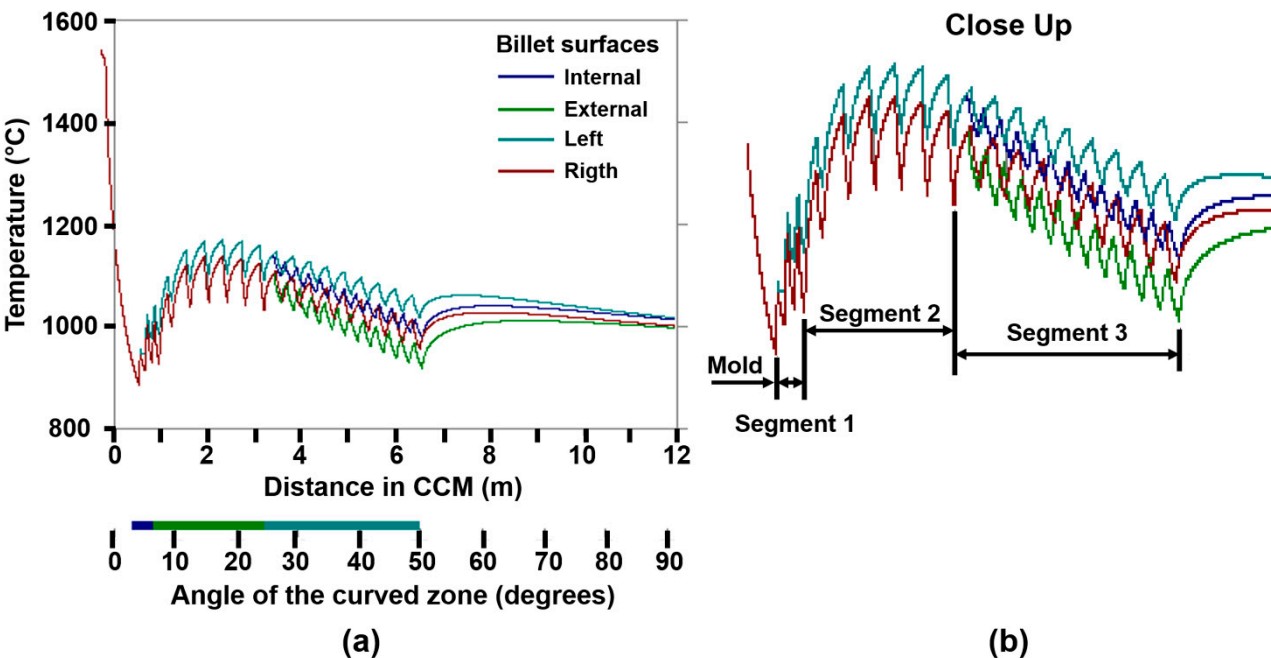

**Figure 8.** (**a**,**b**) Temperature on the billet surfaces due to the symmetrical heat removal applied (consequence of a non-symmetrical spray distribution and differential water flow rates).

During the first two segments, the curve with the highest temperatures are due to the lowest flow rates (left billet surface) and the lowest temperatures are due to the highest water flow rates (right billet surface). Moreover, Figure 8 shows the temperature difference due to the differential spray arrangements along the cast direction in segment 3 of the SCS. There were thirteen and twelve sprays for quenching the internal and external sides but only nine sprays along the left and right billet side surfaces. Moreover, the curves for the internal and external billet surface temperatures rose as continuity lines from the left and right curves, respectively. The maximum temperature difference among the four faces was 130 °C and the maximum rebound temperature was 140 °C, which may have led to thermal cracking.

In the third segment, the water flow rates and number of sprays along the cast direction were different. Then, four curves were displayed in this segment due to the different sprays and water flow rates used to quench each billet surface. Figure 9a–c show the corresponding temperature profiles in the mold. These figures are symmetrical due to the same heat removal conditions applied to the four billet surfaces. The corresponding temperature profiles to the SCS are in Figure 10a–i. Here, the symmetry of the profiles inside the mold is slowly missed. In Figure 10a,b, the circles in the core billet are deformed and the isothermal regions near corners are different and not symmetrical. In Figure 10h,i, the circular isotherms are deformed and moved toward the surfaces with the highest removal heat; furthermore, the isotherms adopt a hyperbolic shape, evidencing the non-symmetrical heat removal pattern, including the liquid core.

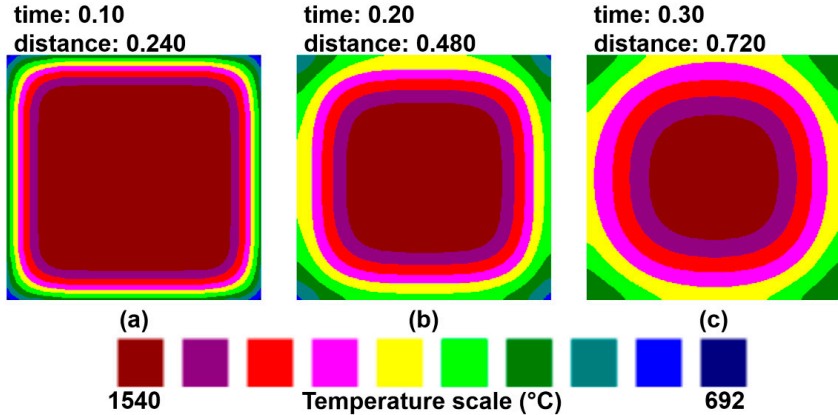

**Figure 9.** (**a**–**c**) Temperature profiles inside the mold (assumed symmetrical removal). The positions of these profiles are a function of the distance, taken as a reference at the meniscus level; time (min) and distance (m).

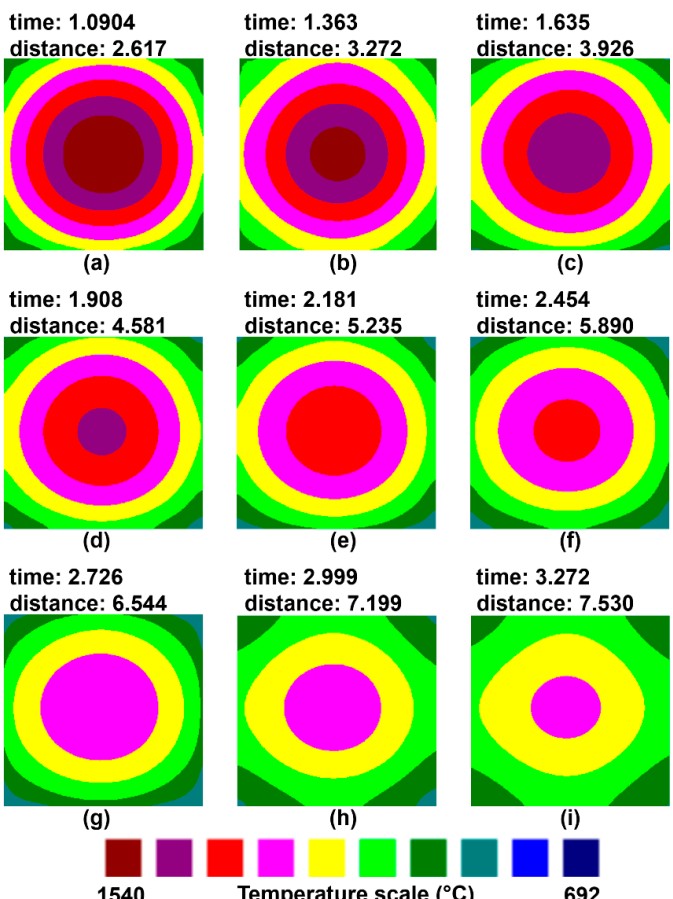

**Figure 10.** (**a**–**i**) Temperature profiles in the SCS. Here, the temperatures' profiles became non-symmetrical due to different water flows rates applied to quench every side; time (min) and distance (m).

Finally, at the end of the SCS, the temperatures of the four surfaces tended to homogenize due to the heat removal by radiation out the SCS, which is less intense, and the heat conducted redistributed inside the billet core. Figure 11a–f show the temperature profiles in the free or radiation region of the CCM. Here, the latent heat inside the billet core was slowly distributed, tending to adopt a homogeneous temperature. Figure 8 shows the thermal conditions of the surface at the end when the billet was driven out of the CCM.

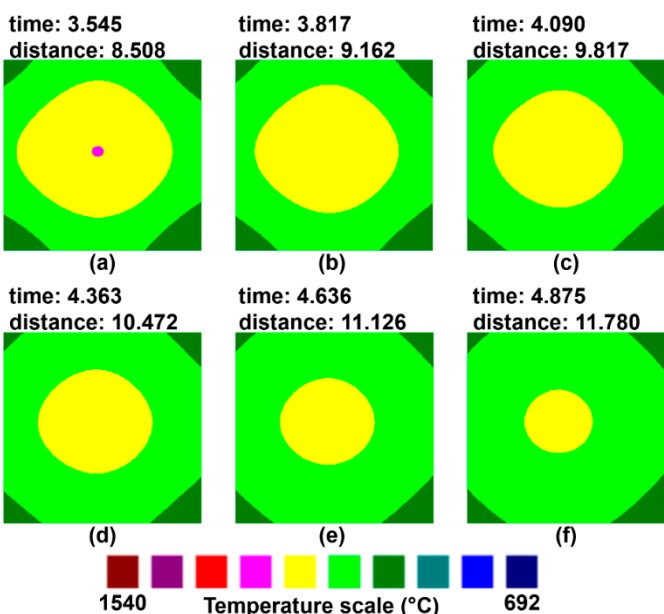

**Figure 11.** (**a**–**f**) Temperature profiles in the free zone; time (min) and distance (m).

The influence over the temperature profiles was more evident near the billet surfaces than in the billet center because of the heat removal conditions applied. The corners were the coldest areas because the perimeter/area ratio was larger than in the middle of the billet surfaces. In other words, for the same area, there was a more extended perimeter in which the heat flux operated. These influences are depicted in Figures 8–11. Although considerable changes in the steel thermal behavior were evident, as in Figures 9–11, sometimes the influence over the solidification profile was not significant due to the small temperature difference between ($T_{Liq}$) and ($T_{Sol}$), depending on the steel chemistry.

*4.2. Case 2*

Figure 12 shows another example for a different layout of a CCM and both the configuration and operating conditions are in Table 5. Figure 13 shows the temperature profiles and curves of surface temperatures. The simulated billet was a square with section of (0.125 × 0.125 m); the cast speed was 2.40 m/min, and the curved radius of the CCM was 7.95 m. The same mesh (200 × 200 nodes) was helpful for the calculation. Figure 13 resembles that shown in Figure 8 as both report the mid-face temperatures of the billets and it is hard to find differences. However, the differences are in the thermal fields inside the billet and in the temperature profiles, along with the billet sides.

**Table 5.** Operation conditions of the SCS (segments 1, 2, and 3), case 2.

| Segment | 1 | | | | 2 | | | | 3 | | | |
|---|---|---|---|---|---|---|---|---|---|---|---|---|
| **Surface** | **Internal** | **External** | **Left** | **Right** | **Internal** | **External** | **Left** | **Right** | **Internal** | **External** | **Left** | **Right** |
| Water flow rate (L/min) | 15 | 15 | 25 | 22 | 12 | 12 | 17 | 15 | 9 | 9 | 12 | 10 |
| Sprays on cast direction | | 8 | | | | 11 | | | | 8 | | |
| Sprays on the lateral direction | 1 | 1 | 1 | 1 | 1 | 1 | 1 | 1 | 1 | 1 | 1 | 1 |
| Nozzle diameter (m) | | | | | | 0.003 | | | | | | |
| $\theta$ | | 7.50 | | | | 22.5 | | | | 30 | | |
| $\Omega$ | | 80 | | | | 60 | | | | 60 | | |
| $D_{bs}$ (m) | | 0.060 | | | | 0.110 | | | | 0.075 | | |

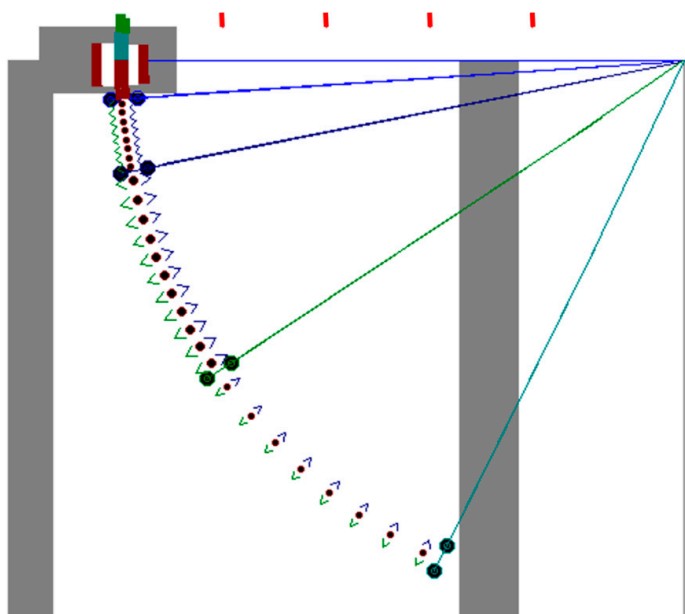

**Figure 12.** Layout of a symmetrical CCM (curved zone), case 2.

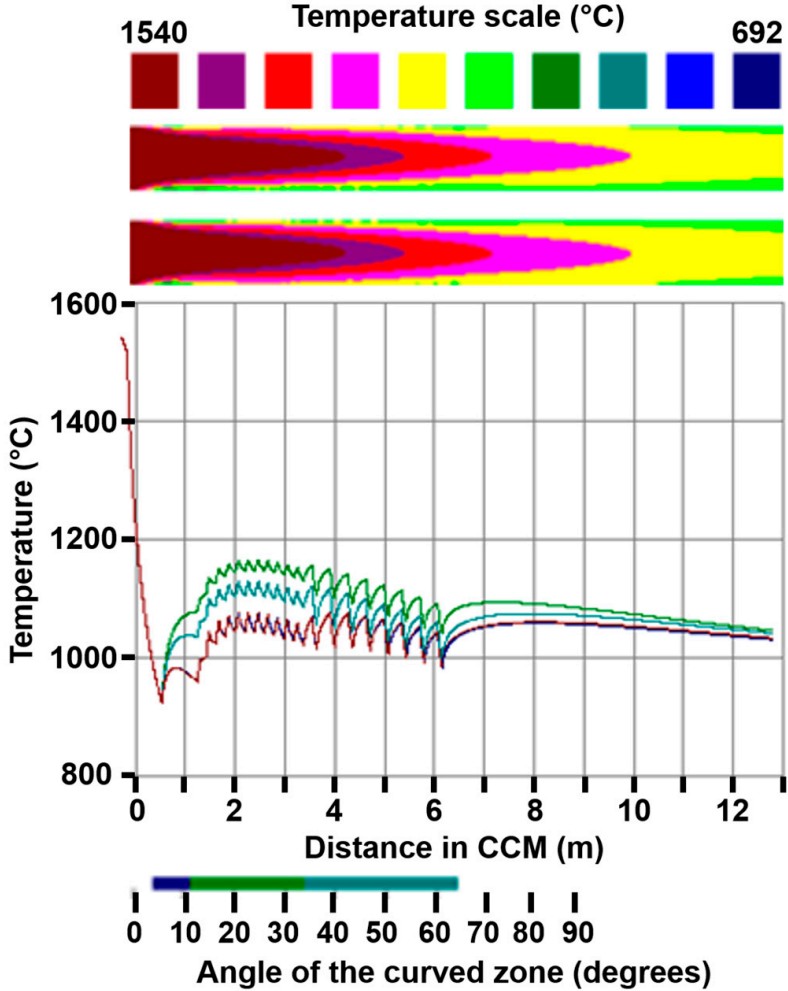

**Figure 13.** Temperature profiles and surface temperature of a squared billet with no homogeneous heat removal.

Here, only one single line represents the temperature when the steel remained inside the mold. Then, three different curves diverged when the billet was in the first segment due to the sprayed areas along the cast direction. When the billet was in the second and third segments, an alternate quenching and reheating behavior appeared due to different heat removal rates. The coldest zones due to corner effects were also affected, showing no homogeneous behavior. Finally, at the end of the curved zone, where no more spray segments quenched the billet surfaces, all the temperatures tended to adopt the same behavior due to heat flowing from the core.

Although the CCM is symmetrical, different surface temperatures resulted after the analysis due to different applied heat fluxes to each billet face. Figure 14 shows a set of perpendicular or top views. Here, it is possible to observe that only one single curve is inside the mold due to applying the same heat removal on the four billet surfaces, as shown in Figure 14a,b. The non-symmetric profiles emerged when the billet was in the SCS from Figure 14c–r. The isothermal regions changed to rotated squared profiles because of the uneven heat removal rates applied at short distances on the billet surfaces. Moreover, some profiles tended to adopt the same hyperbolic form inside the billet core, like those shown in case 1.

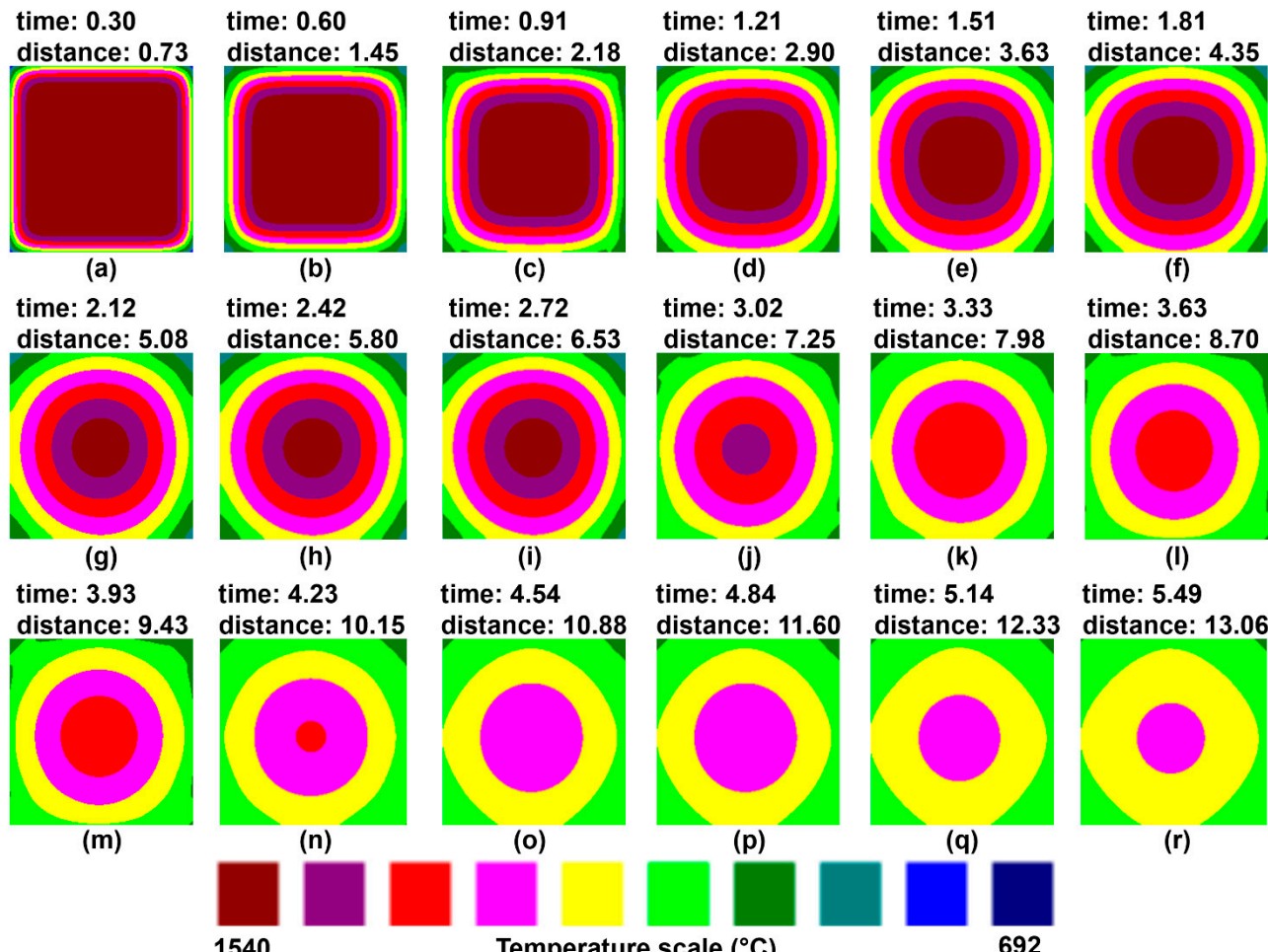

**Figure 14.** (**a–r**) Temperature profiles for steel top views; time (min) and distance (m).

### 4.3. Case 3

Figure 15 shows a layout of a CCM used for casting steel slabs. It is more significant than a billet caster and there are more segments in the SCS because a bigger steel volume needs a longer quenching process. Furthermore, cast speeds are also slower than those for

casting billets due to the same reason. The general casting conditions are in Table 6, while the CCM geometry and operating conditions are in Tables 7 and 8, respectively. Table 7 is divided into two sections to identify the values for the curved and straight zones. Here, (*ds*) is the distance from the nozzle to the slab surface and this is a known data, while (*dw*) and (*dnw*) are calculated in addition to the sprayed and non-sprayed distances. The sub-index (*zn*) is the identification of a segment.

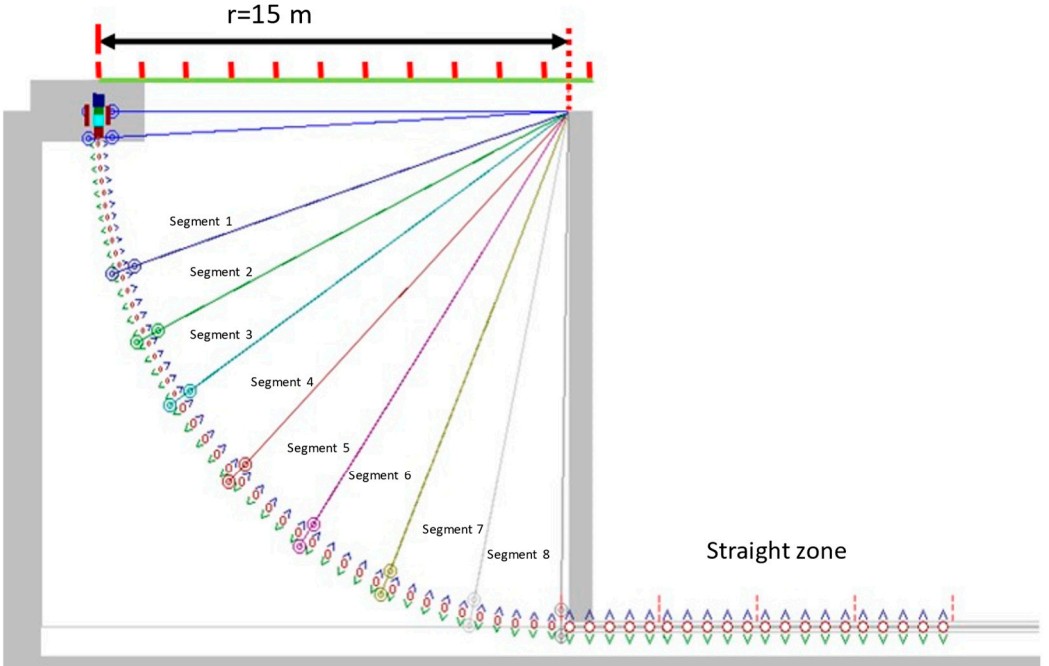

**Figure 15.** Layout of the slab caster.

**Table 6.** General configuration of the CCM for casting slabs.

| Cast Temperature (°C) | $R_C$ (m) | $\theta_0$ | Narrow Side $D_x$ (mm) | Broad Side $D_y$ (mm) | Mold Length (m) | Meniscus Level (%) |
|---|---|---|---|---|---|---|
| 1545 | 10.5 | 23.5 | 200 | 1100 | 1.10 | 82 |

**Table 7.** Dimensions of the SCS in the curve zone (for slabs).

| Zone | $\theta$ | $\Sigma\theta$ | $Rd$ (m) | $\Omega$ | Sprays on Casting Direction | $ds'_{ZN}$ (mm) | $dw'_{ZN}$ (mm) | $dnw'_{ZN}$ (mm) |
|---|---|---|---|---|---|---|---|---|
| **Curved Zone** | | | | | | | | |
| 1 | 12 | 18 | 3.29 | 60 | 11 | 500 | 329 | 171 |
| 2 | 8 | 26 | 4.76 | 55 | 5 | 500 | 467 | 33 |
| 3 | 8 | 34 | 6.23 | 50 | 5 | 500 | 467 | 33 |
| 4 | 11 | 45 | 8.24 | 50 | 5 | 750 | 660.5 | 89.5 |
| 5 | 11 | 56 | 10.26 | 50 | 5 | 750 | 660.5 | 89.5 |
| 6 | 11 | 67 | 12.28 | 50 | 5 | 750 | 660.5 | 89.5 |
| 7 | 11 | 78 | 14.29 | 50 | 5 | 750 | 660.5 | 89.5 |
| 8 | 10 | 88 | 16.30 | 50 | 5 | 750 | 660.5 | 89.5 |

**Table 7.** *Cont.*

| Straight Zone | | | | | | |
|---|---|---|---|---|---|---|
| Zone | *Rd* (m) | Ω | Sprays | $ds'_{ZN}$ (mm) | $dw'_{ZN}$ (mm) | $dnw'_{ZN}$ (mm) |
| 9 | 18.61 | 55 | 5 | 750 | 630.4 | 119.6 |
| 10 | 20.92 | 55 | 5 | 750 | 630.4 | 119.6 |
| 11 | 23.22 | 55 | 5 | 750 | 630.4 | 119.6 |
| 12 | 25.52 | 55 | 5 | 750 | 630.4 | 119.6 |

**Table 8.** Operating conditions of the SCS in the curve zone (for slabs).

| Zone | Sprays on the Lateral Direction | Nozzle Diameter (mm) | Ω | $D_{bs}$ (mm) |
|---|---|---|---|---|
| 1 | 5 | 2.5 | 40 | 180 |
| 2 | 5 | 2.5 | 40 | 180 |
| 3 | 4 | 2.5 | 45 | 150 |
| 4 | 4 | 2.5 | 45 | 150 |
| 5 | 3 | 3 | 30 | 120 |
| 6 | 3 | 3 | 30 | 120 |
| 7 | 3 | 3 | 30 | 120 |
| 8 | 3 | 3 | 30 | 120 |
| 9 | 3 | 3 | 30 | 120 |
| 10 | 3 | 3 | 30 | 120 |
| 11 | 3 | 3 | 30 | 120 |
| 12 | 3 | 3 | 30 | 120 |

Figure 16 shows the left slab cross-sections from the meniscus level to the end and the temperature profiles, which are the same as the right cross-sections due to symmetry-related reasons. It is possible to observe that the general heat removal rate increases when the slab casting speed is slow due to longer residence times under cooling effects in the SCS. Moreover, it is more complicated to appreciate the influence of different heat removal conditions due to the intensity of the heat flux conducted from the big slab core.

Figure 17a,b show the effects of the casting speed on the surface temperature for the narrow and broad faces of the slab. Here, the difference between the surface temperature for both simulations is notorious. The slab casting speed at 1.0 m/min ran slowly and the heat removal was much more intense than the slab cast at 1.30 m/min. Furthermore, the surface temperatures for narrow and wide slab surfaces were also higher, as seen in Figure 17b, but the difference between the minimum and maximum temperature on the SCS was smaller due to the heat conducted from the slab core. Tables 9 and 10 show the differences between the actual temperatures measured directly using a digital pyrometer and the temperatures simulated computationally, showing the lowest and highest temperatures. These correspond inversely to the highest and lowest water flow rates applied.

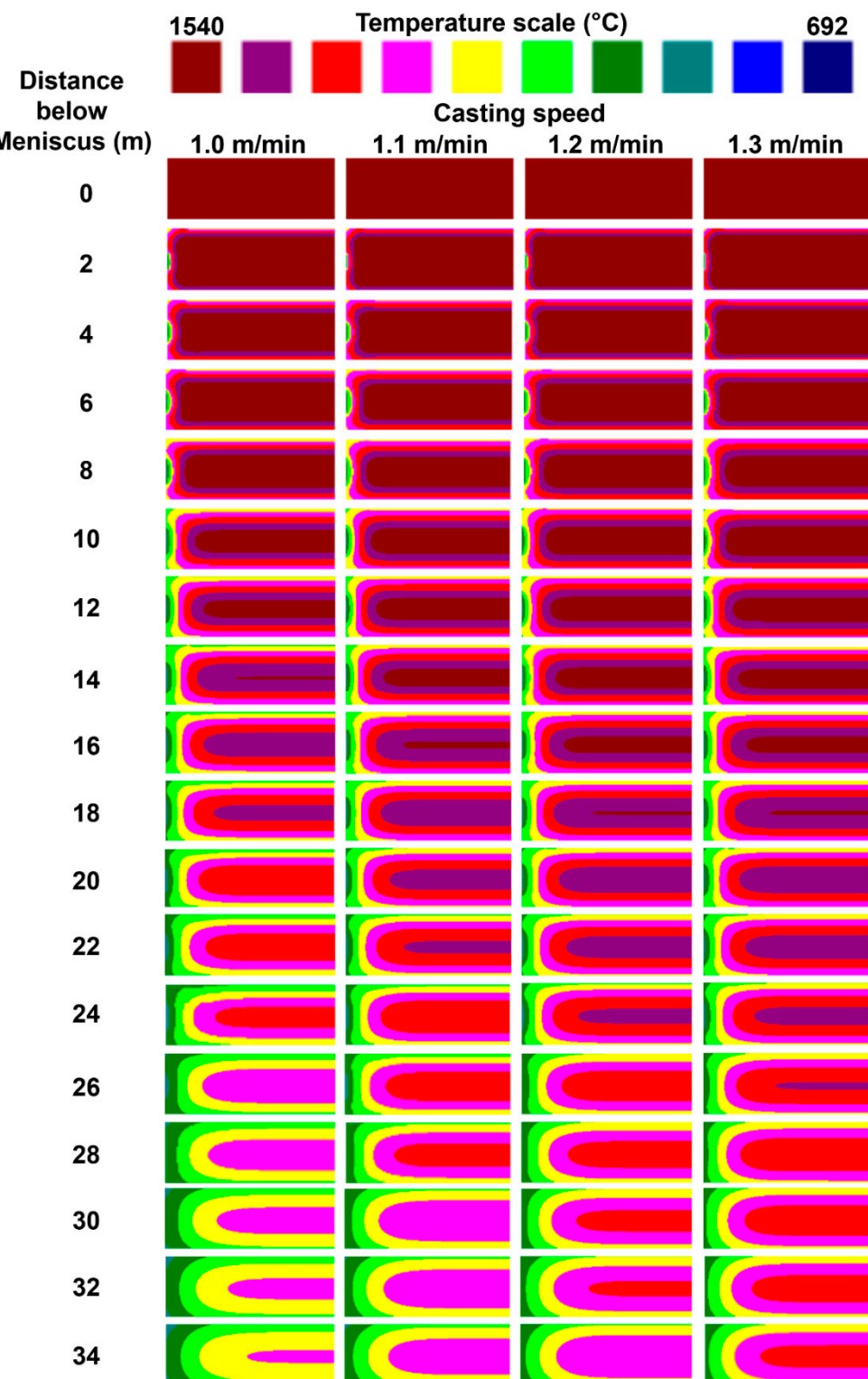

**Figure 16.** Temperature profiles calculated computationally for case 3 considering a slab cast running at four different cast speeds (perpendicular views to the cast direction for the slabs).

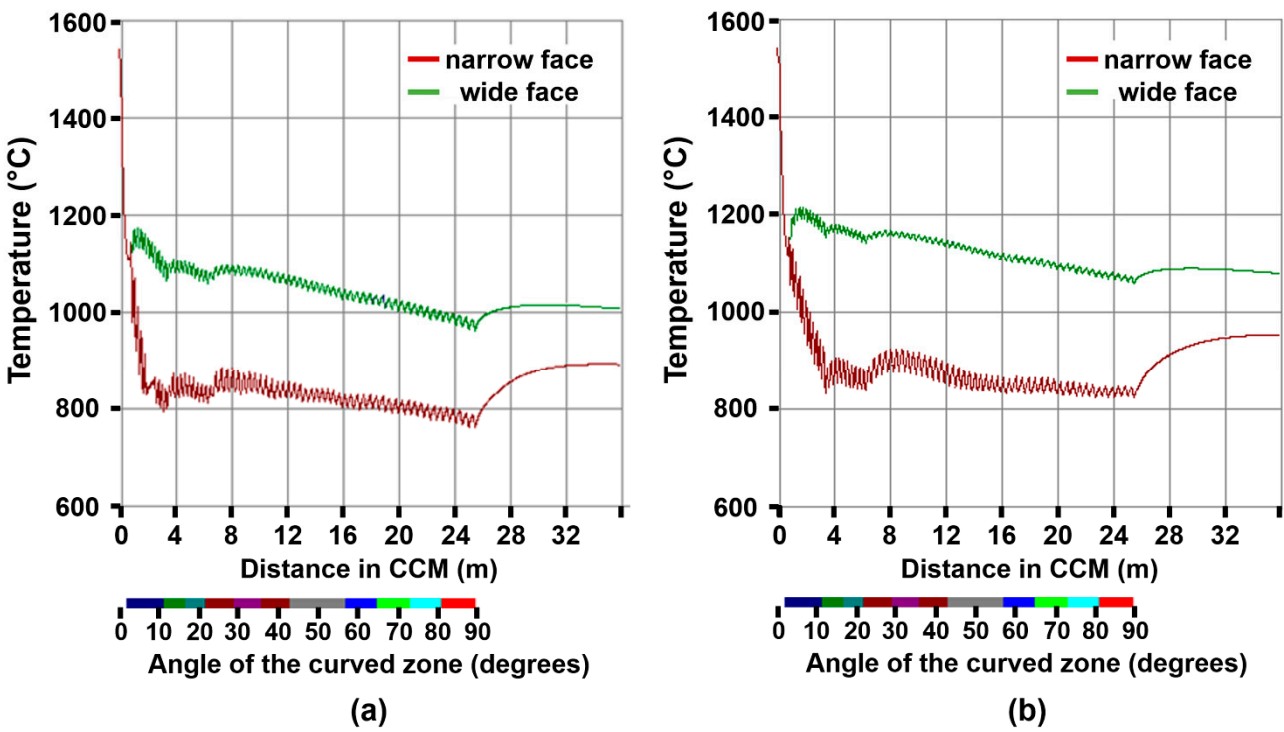

**Figure 17.** Surface temperatures on the narrow and wide slab faces (**a**) for a casting speed = 1.0 m/min and (**b**) 1.30 m/min.

**Table 9.** Calculated errors for the simulated billets, case 1.

| Distance below Meniscus (m) | Temperature Measured (°C) | Temperature Simulated (Lowest) 200 × 200 Mesh (°C) | ΔT (°C) | Temperature Simulated (Highest) 200 × 200 Mesh (°C) | ΔT (°C) |
|---|---|---|---|---|---|
| 2.25 | 1072 | 1065 | −7 | 1080 | 8 |
| 3.75 | 1111 | 1070 | −41 | 1125 | 14 |
| 4.8 | 1080 | 1050 | −30 | 1098 | 18 |
| 5.0 | 1054 | 1017 | −37 | 1089 | 35 |
| 5.4 | 1050 | 1020 | −30 | 1075 | 25 |
| 7.0 | 1035 | 995 | −40 | 1052 | 17 |
| 7.5 | 1033 | 1002 | −31 | 1045 | 12 |
| 8.0 | 1024 | 1010 | −14 | 1037 | 13 |
| 8.5 | 1024 | 1012 | −12 | 1031 | 7 |
| 9.0 | 1022 | 1012 | −10 | 1025 | 3 |
| 9.5 | 1025 | 1010 | −15 | 1022 | −3 |
| 10 | 1015 | 1006 | −9 | 1017 | 2 |
| 10.5 | 1010 | 1004 | −6 | 1014 | 4 |

**Table 10.** Calculated errors for the simulated billets, case 2.

| Distance Below Meniscus (m) | Temperature Measured (°C) | Temperature Simulated (Lowest) 200 × 200 Mesh (°C) | ΔT (°C) | Temperature Simulated (Highest) 200 × 200 Mesh (°C) | ΔT (°C) |
|---|---|---|---|---|---|
| 1 | 975 | 920 | −55 | 1040 | 65 |
| 2.5 | 1095 | 1040 | −55 | 1140 | 45 |
| 3.0 | 1095 | 1030 | −65 | 1135 | 40 |
| 4.1 | 1090 | 1025 | −65 | 1120 | 30 |
| 4.9 | 1056 | 1001 | −55 | 1108 | 52 |
| 5.5 | 1045 | 980 | −65 | 1099 | 54 |
| 6.0 | 1035 | 976 | −59 | 1084 | 49 |
| 6.5 | 1031 | 970 | −61 | 1068 | 37 |
| 7.0 | 1035 | 980 | −55 | 1095 | 60 |
| 7.5 | 1030 | 962 | −68 | 1072 | 42 |
| 8.0 | 1048 | 995 | −53 | 1051 | 3 |
| 8.5 | 1025 | 1001 | −24 | 1047 | 22 |
| 9.0 | 1025 | 1003 | −22 | 1041 | 16 |
| 9.5 | 1020 | 1002 | −18 | 1034 | 14 |
| 10 | 1021 | 999 | −22 | 1028 | 7 |

None of the measured temperatures was out of the range of those calculated numerically. Thus, it is possible to affirm that the algorithms gave a promising approach for the steel thermal behavior. The temperature differences are prominent for the second case due to the CCM geometry and the operating conditions were notoriously different, generating a significant amplitude between the minimum and maximum heat removal applied.

The temperatures in Table 11 correspond to the analysis of case 3 for a slab with a cast speed of 1.0 m/min. Due to the complexity of the CCM, it is possible to take some measurements for the straight zone. The temperatures corresponded to the internal slab face. These are remarkably similar in comparison with those calculated computationally; thus, it is possible to confirm the values obtained computationally as trustworthy.

**Table 11.** Calculated errors for the simulated slabs, case 3.

| Distance below Meniscus (m) | Temperature Measured (°C) | Temperature Simulated 550 × 100 Mesh (°C) | ΔT (°C) |
|---|---|---|---|
| 24 | 800 | 810 | 10 |
| 26 | 795 | 807 | 12 |
| 28 | 850 | 842 | −8 |
| 30 | 860 | 851 | −9 |
| 32 | 882 | 874 | −8 |
| 34 | 890 | 883 | −7 |
| 36 | 886 | 881 | −5 |

## 5. Conclusions

A numerical algorithm designed to predict the thermal fields of continuous casting machines for billet and slabs is presented in this work and the conclusions derived from the simulation results are as follows:

1. The direct approach of calculating the heat transfer coefficients through the appropriate dimensionless numbers, rather than through other reported empirical correlations, is suitable to predict the temperature fields in slab and billet machines.
2. The surface temperatures along the casting length of slabs and billets using this algorithm match acceptably well the temperature measurements.
3. The matching between the measured temperatures and those simulated indicate that the mesh size of $200 \times 200$ nodes is large enough to obtain reliable thermal predictions.
4. The algorithm is versatile as it permits the friendly changes of different casting machines, including the use of different types of water spray nozzles.

**Author Contributions:** Conceptualization, A.R.-L.; methodology, A.R.-L., R.D.M. and A.N.-B.; software, J.R.-Á. and C.R.M.-V.; formal analysis, A.R.-L., R.D.M. and A.N.-B.; investigation, A.R.-L. and O.D.-M.; supervision, A.R.-L. and R.D.M. All authors have read and agreed to the published version of the manuscript.

**Funding:** This research study was funded by Consejo Nacional de Ciencia y Tecnología (CoNaCyT), SNI, Instituto Politécnico Nacional (IPN), and Universidad Autónoma de Coahuila (UAC).

**Institutional Review Board Statement:** Not applicable.

**Informed Consent Statement:** Not applicable.

**Acknowledgments:** The authors wish to thank the Autonomous and Technological Institute of Mexico (ITAM), Consejo Nacional de Ciencia y Tecnología (CoNaCyT), and Instituto Politécnico Nacional (IPN) for their technical and financial support.

**Conflicts of Interest:** The authors declare no conflict of interest.

## Nomenclature

| | |
|---|---|
| AR1 | upper austenite-ferrite transformation temperature |
| AR3 | lower austenite-peralite transformation temperature |
| $C_P$ | heat capacity |
| D | diameter |
| h | heat transfer coefficient |
| H | enthalpy |
| k | thermal conductivity |
| l | billet side length |
| Nu | Nusselt number |
| q | heat flux |
| Pr | Prandtl number |
| Re | Reynolds number |
| T | temperature |
| t | time |
| W | mass of steel |
| Greek symbols: | |
| $\alpha$ | thermal diffusivity |
| $\Delta x$, $\Delta y$, and $\Delta z$ | increments of distance and time |
| $\mu$ | viscosity |
| $\Omega$ | shooting angle of every spray |
| $\rho$ | density |
| $\theta_0$ | the first angle of the secondary colling system |
| Subindexes: | |
| bs | length of billet side cooled by the spray |
| i,j | nodes in the computational mesh |
| Liq | liquidus temperature |
| m | mold |

| mushy | two-phase region of solidification depending on the steel chemistry |
| n and side | indexes to indicate that these values can differ for every segment of the SCS and every billet side |
| s | water spray |
| sol | solidus temperature |
| w | water |
| Acronyms: | |
| CCP | continuous casting process |
| PCS | primary cooling system, i.e., the mold |
| SCS | secondary cooling system, i.e., the water spray segments of a machine |
| CCM | continuous casting machine |
| TCO | casting temperature at the meniscus level |
| RC | machine radius |

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
