# Peer review of "Analysis of Non-Symmetrical Heat Transfers during the Casting of Steel Billets and Slabs"

_metals, doi:10.3390/met11091380_

Round 1
Reviewer 1 Report
Comments and specific questions requiring clarification (in addition to the comments in the text of the article, in a pdf file):
- Computational Representation of Steel Casting.1. Explain shortly for me please why the model use the finite difference method? Why not for example finite element method or other numerical method?
- Heat Removal and Conduction Inside the Billet Core.
1. If it is possible use additional explanation of symbols used in equations – bellow the equations – it refers to all equations in manuscript
Simulations and results.
- Based on the results presented on the figure no. 8 explain shortly what is the differences (in °C or in %) between measured points in this case. Based on the data presented on the figure no 8 it it difficult for me to read this information. Explain briefly what possible temperature differences could have an effect on. Maybe cracking, but it's probably not in this temperature range.
- Results presented on the figure no 8 and 13 seems to be similar – can You explain it shortly for me – please?
- Write something more about measurement with using digital pyrometer please – especially I am interesting what was the value of emmisivity. How the emmisivity was determined at this temperature range and how it was changed? Did You use one value of emmisivity or the emmisivity value was used as depended of temperature?
General comments (suggestions):
- Please use full name of the used shortcut first – then You can use only shortcut. Sometimes in the manuscript You use shortcut without full explanation what does it mean. It is probably known for specialists in this scientific area but for others readers will be more clear if You explain used shortcuts first.
- If it is possible use additional explanation of symbols used in equations – bellow the equations – it refers to all equations in manuscript
- Add references to the pictures when they are not Your own pictures please – check it please.
- Maybe it will be good idea to add list of used shortcut and symbols description – at the begining of the manuscript – because there are a lot of them in the paper?

Author Response
Dear Professor, here follow our answers to your questions.
- The use of finite difference method instead of the finite element is our a better ability and experience in using and programming the first.
- Yes Sir, Madam, we include a complete nomenclature of symbols.
- The maximum differences of temperature among the billet surfaces is 130 degrees. The maximum, rebound temperature is 140 degrees. Indeed. thermal stresses can be high. We mention this fact in the corrected manuscript.
- We used a fixed emissivity of 0.85. The reason of using a fix temperature is due to the existence of a oxide layer along the strand.
-
The Figure 13 resembles that shown in Figure 8 as both report the mid-face temperatures of the billets it is hard to find differences. However, the differences are in the thermal fields inside the billet and in the temperature profiles along with the billet sides.
- About all your five suggestions below, we have followed them in the new manuscript.
On the name of my co-authors and me I give you the thanks for your valuable observations on our paper. Thank you very much.
Reviewer 2 Report
The ABSTRACT section is well-presented, relevant for this subject in area of current automation of steelmaking processes and the continuous casting. The aims and objectives of the research are well defined.
The paper is structured properly (INTRODUCTION, MATERIAL & METHODS, RESULTS & DISCUSSIONS, CONCLUSIONS, REFERENCES, etc.).
The INTRODUCTION section provide the necessary background information regarding the continuous casting process & machine, including the transfer of steel-ladle to a tundish, heat transfer stages (PCS, SCS, and the radiation zone) and simulation processes. The section is well-documented.
The METHODOLOGY is relatively well described and the mathematical terms in the model and the assumptions are well-presented. Heat removal (steel is under a sprayed zone, respectively under a no-sprayed area) are large presented, including the different arrangements of billet & slab cooling sprays in continuous casting machines. The Process Simulation and the specific Operating conditions and assumptions, different solidification rates, are well-presented.
The body of paper describe the important RESULTS of the research. The SIMULATIONS AND RESULTS are presented in several cases. The Tables are representative and the Figures & Graphs have good qualities.
The CONCLUSION section succinctly summarize the major points of the paper, quite ambiguous, but presented concisely and to the point.
Author Response
Dear Professor,
Thanks for your encouraging comments. We appreciate your observations.
Reviewer 3 Report
With the title of Analysis of Non-Symmetrical Heat Removal During Casting of Steel Billets and Slabs, the authors make a complex study about the heat removal during continuous casting process with the help of computation tools. In my opinion the objective of the paper is not easy, so every effort made in this way are welcome. Congratulations to authors for the work.
From my point of view the work is too long, with a lot of figures to be studied (some figures have to many figures inside them) and tables (too many data numbers too). In occasions the work is difficult to read because not all the acronyms, abbreviations or symbols used are well described in the text. Some of then are described two pages later from their original place in the work. With all due respect to authors, I would like to give a little advice: all the symbols-acronyms-abbreviations used must be identified clearly in the work (even when a symbol, for example, can be obvious) the first time that they are used (inside the text, in an equation, figure or table). Chemical symbols are excluded from my previous advice. Maybe this work needs a list with all the symbols/abbreviations/acronyms used on it (to make reading easier).
In Figs 8 and 13: there are temperature data for CCM < 0; is that ok?
Nothing else on my part.
Best regards!
Author Response
Dear Professor,
Yes, looking throughout it, this paper looks and is heavy to read. Especially your observations about the acronyms and abbreviations. In the new manuscript, we try to make it easy the reading following your suggestions.
On the name of my co-authors and me, I give you the thanks for going into the tough task and review of our paper.
Best regards